# The Lockdown Impact on the Relations between Portuguese Parents and Their 1- to 3-Year-Old Children during the COVID-19 Pandemic

**DOI:** 10.3390/children9081124

**Published:** 2022-07-28

**Authors:** Frederica Vian, Rita Amaro, Sofia Vaz Pinto, Henrique de Brito, Raissa Rodrigues, Rita Rapazote, Pedro Caldeira da Silva, Marta Alves, Ana Luisa Papoila

**Affiliations:** 1Department of Child and Adolescent Psychiatry, Centro Hospitalar Universitário de Lisboa Central, 1169-045 Lisbon, Portugal; aritaamaro@gmail.com (R.A.); ana.sofia.pinto@campus.ul.com (S.V.P.); henriquebrito@campus.ul.com (H.d.B.); raissa.velasco.rodrigues@gmail.com (R.R.); 2Centro de Estudos do Bebé e da Criança, Centro Hospitalar Universitário de Lisboa Central, 1169-024 Lisbon, Portugal; anarita.rapazote@gmail.com (R.R.); pedrocds@gmail.com (P.C.d.S.); 3Reserch Centre, Epidemiology and Statistics Unit, Centro Hospitalar Universitário de Lisboa Central, 1169-045 Lisbon, Portugal; marta.l.alves@gmail.com (M.A.); apapoila@hotmail.com (A.L.P.); 4CEAUL (Centre of Statistics and its Applications), Lisbon University, 1499-002 Lisbon, Portugal; 5NOVA Medical School, Universidade Nova de Lisboa, 1169-056 Lisbon, Portugal

**Keywords:** COVID-19 pandemic, lockdown impact, parenting, children, resilience

## Abstract

Many countries have applied mandatory confinement measures in response to the COVID-19 pandemic, such as school and kindergarten closures, which confined families to their homes. The study concerns the impacts of the first COVID-19 lockdown on the relationships between Portuguese parents and their children, in a non-clinical population composed of fathers and mothers of children between the ages of 12 months and 3 years and 364 days. An online questionnaire (set by the research team) and the Parenting Daily Hassles Scale (PDHS) concerning the confinement period were applied between 17 June and 29 July 2020. To assess the impacts of the lockdown, outcomes regarding the impacts perceived by the parents, the potential regression in the development of children, and the willingness to promote changes in family routines in the future, were considered. Of the total sample (*n* = 1885), 95.4% of the parents (*n* = 1798) said that, after confinement, the relationship with their children had improved or remained similar to the pre-confinement period; 97.3% (*n* = 1835) noticed positive changes in the development of their children, and 63.7% (*n* = 1200) noted that the relationships with their children during the confinement period would lead to some changes in family routines in the future. Multivariate regression analyses showed that most of the sociodemographic variables chosen were not associated with the outcomes. However, significant levels of pressure over parenting and parental overload (reported by high scores in the PDHS intensity and frequency scales), challenging behaviors of the children, and the impacts they had on parental tasks had negative influences on the studied outcomes. On the contrary, the number of adults living with their children, the perceptions regarding the development of their children, and sharing new experiences with them were significant factors for positively-perceived impacts on the relationships between them or in the desire to bring about changes in family routines in the future. The impacts of the lockdown on the relationships between parents and children aged between 1 and 3 years old were more dependent on relational aspects and on the parents’ sense of competence in exercising parental functions. We conclude that, despite the increased demands imposed by the lockdown, nearly all of the parents evaluated the quality of the relationship with their children as positive after this period.

## 1. Introduction

On 2 March 2020, the first two cases of coronavirus disease 2019 (COVID-19) were announced in Portugal [1]. At the time, the disease had already spread across the world, affecting 90,000 people and killing over 3000 [2]. Even before the World Health Organization (WHO) announced that the world was facing a global pandemic, some countries (such as Portugal) enforced exceptional measures to contain the spread of the virus [3]. 

A state of emergency was declared by the Portuguese government almost immediately after the first cases were diagnosed. A national lockdown followed, involving the closures of kindergartens, schools, universities, social and recreational venues, and most commerce. All sports and public events, religious gatherings, and public assemblies were forbidden; rules involving social distancing and working from home were implemented. The routines of all Portuguese families (as with many other countries) deeply changed overnight. 

The educational role of parents at home had never been more important [4]; it involved parents looking after their children, home-schooling (almost on their own), and facing all of the challenges of working from home while running their households.

Portugal managed to escape the tragic events of other southern European countries, such as Spain and Italy, although a considerable number of families still had to deal with the dreadful consequences of the spread of the virus and its death toll. In Italy, for example, studies in an adult population showed that COVID-19 quarantine in April 2020 was associated with moderate-to-severe stressful impacts [5]. The Portuguese healthcare system barely managed to hold on and the already fragile and unstable economy suffered.

In Portugal, during the first lockdown, with few exceptions, families with children under 12 years old were given a choice. One parent could suspend his/her professional activity temporarily and receive financial support from the National Social Security System [6]. Most families, especially those with little children, chose this option.

These exceptional circumstances highly contrasted with what usually took place with young children in Portugal. Data from 2019 [7] showed that, on average, Portuguese children under 3 years old spent 39.1 h weekly with childcare providers—one of the highest rates in the former EU-28 countries [8] (in which the average was 27.4 h). This is a source of concern for early childhood mental health professionals in Portugal.

In the context of the current pandemic, the published studies on the lockdown’s impact on the mental health of the population confirmed that this has been particularly stressful for most families [9,10,11]. 

For instance, one study [4] showed that due to the increased levels of stress in parents (irritability and anxiety) and their difficulty in enjoying the constant interactions with their children, the lockdown impacted the emotional and behavioral problems experienced by minors between the ages of 2 and 14. Since children had their own emotional and behavioral issues to begin with, parents in turn had a harder time coping with the lockdown [9]. 

In another study, conducted to examine parents’ perceived stress and whether reports of rewards and challenges during the lockdown impacted their stress levels, the authors concluded that lockdowns are not uniformly or consistently negative experiences for parents [12]. 

Curiously, some factors (e.g., living in a more at-risk contagion zone or being in closer contact with the effects of the virus) did not relevantly affect the well-being of parents and children in a lockdown context [4,13]. Similarly, the quality of the environment (e.g., the physical characteristics of the living space) was not associated with the psychological hardships of the parents and children. 

Few studies have focused on the impacts of lockdown experiences in families with children who are in early childhood. As we know, parenting in the first three years of a child’s life requires healthy parental involvement, affectionate relationships, and parental sensitivity to the child’s needs [14]. Parents who can provide a safe, welcoming, and effective environment, regardless of circumstances, contribute to the healthy social development of their children [15]. In this process, it is paramount that parents spend time with their children to emotionally bond with them [15].

In this context, regarding Portuguese families, we thought some questions were worth asking: was the lockdown a unique opportunity for Portuguese parents to be with their babies? How did families adapt to this lockdown context? What were the parental perceptions on how the lockdown impacted their relationships with their babies? What were their perceptions on the impacts that this highly intensive relational experience had on the development of their babies? Once the lockdown ended, what was the potential that this experience would result in permanent changes in the routines of Portuguese families?

With this in mind, the purpose of this study was to look at the impacts that the lockdown had on the relationships between Portuguese parents and children between the ages of 1 and 3 during the first COVID-19 lockdown.

## 2. Materials and Methods

A convenience sample involving parents of children between 1 and 3 years old was obtained through a web-based cross-sectional survey. Eligibility criteria included caregivers of children who were 1, 2, or 3 years of age, who lived in Portugal, and who agreed to take part in the study after completing an informed consent request. 

The online questionnaire was distributed via e-mail and shared via social media for a limited time of 6 weeks (from 17 June to 29 July 2020). It took approximately 15 min to complete, after reading the written consent form and explicitly agreeing to take part in the study. In the case of multiple children, the parent was asked to report on one child only. Those who did not fully complete the questionnaire or refused to provide consent were automatically excluded. By participating in the survey, participants understood that all of the information obtained would be strictly confidential, with guaranteed anonymity and data protection in future publications on the topic.

The survey was comprised of three parts. In the first section, participants were asked to report their social demographic data and answer some questions about family routines before and during the lockdown period. In the second section, we assessed the frequency and intensity/impact of 20 experiences that could be a ‘hassle’ to parents during the lockdown period by using the Parenting Daily Hassles Scale (PDHS) [16,17]. In the last part of the survey, we assessed parental perceptions about the impacts of this period on the parents’ relationships with their children. Participants were allowed to leave free comments at the end.

Accordingly, sociodemographic characteristics were obtained, including gender, age, place of residency, marital status, household composition, employment status, and medical history that included past and/or current psychiatric disorders. Regarding family routines before and during the lockdown, participants were asked to provide information about who took care of the children (if they stayed at home, were in kindergarten, or were cared for by someone else) and how many hours they spent with them before the lockdown. They were also asked how the routines changed during the lockdown: if there were changes in the household composition and employment status and who took care of their children during this period (if they were their main caregivers or if they shared this task with someone else). 

The PDHS scale is used in a wide variety of research studies concerning children and families; it is used to assess the frequency and intensity/impact of 20 experiences that can be a ‘hassle’ to parents, such as “*Continually cleaning up messes of toys or food*” or “*The kids interrupt adult conversations or interactions*”. 

We used this scale to determine the intensity/impact surrounding the typical daily hassles of parenthood. Caregivers typically enjoy completing the scale because it touches on aspects of being a parent that are very familiar. It helps them express (1) what it feels like to be parents and (2) their need for help with parenting. In this way, our main intention was to make them reflect on these questions before asking them how the lockdown impacted their relationships.

The caregiver was asked to score each item in two different ways for frequency (range 0–80) and intensity (range 0–100). The frequency of each item provides an ‘objective’ marker of how often it occurs. The intensity or impact score indicates the caregiver’s ‘subjective’ appraisal of how much those events affect or ‘hassle’ them. The time frame for this scale can vary according to the focus of the assessment. In our survey, we asked parents to focus on the lockdown period. The scale can be used in two distinct ways: (a) via the total of the frequency and intensity scales can be obtained, or (b) by considering the two subscales “challenging behavior” (sum of the items 2, 4, 8, 9, 11, 12, and 16; range 0–35) and “parenting tasks” (sum of the items 1, 6, 7, 10, 13, 14, 17, and 20; range 0–40) derived from the intensity scale. There is no cut-off for any of the scales but total scores above 50 on the frequency scale or above 70 on the intensity scale indicate, on the one hand, a high frequency of potentially hassling happenings, and on the other hand—that the parent is experiencing significant pressure in parenting [17].

Participants were asked about how their relationships with their children changed after the lockdown period (did it become better, stay the same, or become worse). They were also asked about the developments or regressions of their children, changes in behavior, the level of interference in carrying out daily tasks (due to the child’s behavior), how they would rate the child’s behavior compared with the period prior to the lockdown and, finally, if the relational experience would promote, in the future, any change in their family’s routine. From this information, three questions were chosen as outcomes that best defined the impact of this period, concerning the parents’ relationships with their children, namely the impact perceived by parents, the potential regressions in the development of their children, and willingness to promote some change in the family routines in the future. 

An exploratory analysis of the variables under study was carried out with categorical variables described by frequencies (percentages), and the remaining variables by the means (standard deviation; minimum, maximum). 

Univariate and multivariate logistic regression models were applied to identify the variables that might have been associated with how the lockdown impacted the relationships between parents and children between the ages of 1 and 3, defined by the three outcomes previously mentioned. Demographic characteristics and family routines before, during, and after the COVID-19 lockdown were considered in this analysis as independent variables. 

For the multivariate models, all variables in the univariate analysis that attained a *p*-value ≤ 0.25 were selected. Adjusted odds ratios (OR) were estimated with corresponding 95% confidence intervals (95% CI).

A level of significance α = 0.05 was considered. Data analysis was performed using Statistical Package for Social Sciences (SPSS) software version 25.0 (IBM Corp., released 2017. IBM SPSS Statistics for Windows, Version 25.0. Armonk, NY, USA: IBM Corp.).

## 3. Results

### 3.1. Sociodemographic Characteristics 

The final sample was composed of 1885 caregivers living in Portugal, 91.6% (*n* = 1727) were females. The mean age of the participants was 35.46 years (SD = 4.59; min. 20 years, max. 63 years). Most of the participants were married or in civil unions (92.6%; *n* = 1746) and had university education (86.2%; *n* = 1624) or high school (11%; *n* = 207). Regarding psychiatric history, only 5.8% (*n* = 110) were currently being followed-up in psychiatry or psychology consultations and 27.5% (*n* = 518) reported having been in the past. 

Concerning the ages of children—39.3% (*n* = 740) of the participants were responsible for a child aged 2 years old, 32.6% (*n* = 614) for a child aged 3, and 28.2% (*n* = 531) for a child aged 1. We found that on average the children were in the care of their mothers during maternity leave, for 8.44 months (SD = 6.19; min. 1 month, max. 48 months). A total of 28.9% (*n* = 544) of the children were in their mothers’ care until 5 months of age, 26.2% (*n* = 494) until 6 months, 23.1% (*n* = 435) until 9 months, and 9.5% (*n* = 179) until 12 months (Table 1).

### 3.2. Family Routines before the Lockdown 

Before the lockdown, 85.5% (*n* = 1611) of the children lived with two adults. 51.7% (*n* = 975) of the households had one minor (any person under 18 years old), 35.6% (*n* = 671) had two, and 12.8% (*n* = 239) had three or more.

Within the population covered by this study, 92.6% (*n* = 1746) of the adults worked before the lockdown, 83.4% (*n* = 1573) of the children spent most of their time in a daycare or with a nanny, 8.7% (*n* = 164) were taken care of by their grandparents or other relatives, and 7.9% (*n* = 148) were cared for by their mother, father, or both parents. 

Regarding waking-up hours, bedtime, and pick-ups from daycare schedules, we estimated that each participant in the survey spent, on average, 5.25 h (SD = 1.24; min. 1 h, max. 14 h) with their children (Table 1). 

### 3.3. Family Routines during Lockdown 

Most families stayed at their home residences during the lockdown (87.9%; *n* = 1656), and their households stayed the same (90.2%; *n* = 1700). During the lockdown, 45.5% (*n* = 858) changed their workplaces and worked remotely, 24.7% (*n* = 466) suspended their professional activities, and 13.3% (*n* = 251) remained at their usual workplaces. Regarding taking care of the children, 27.1% (*n* = 510) of the survey participants claimed to be the sole caregivers of their children, and the remaining shared the responsibility with other adults (household companions, other relatives, nannies, or daycare). 

Independent of these factors, during the lockdown, 92.7% (*n* = 1747) of the survey participants claimed that they were able to share new experiences with their children (Table 1).

### 3.4. Parenting Daily Hassles Scale (PDHS) 

Considering the frequency scale, which ranged from 0 to 80, there was a mean total of 45.58 (SD = 8.86; min. 23, max. 78). For approximately 72% (71.7%; *n* = 1352), daily hassles did not constitute an overload (scores lower than 50). Regarding the intensity scale, which ranged from 0 to 100, there was a mean total of 52.49 (SD = 15.23; min. 19, max. 100). In this case, 12% (*n* = 227) of the parents were under high-level pressure (with scores higher than 70).

Regarding the subscales—in the challenging “behavior total” score, a range of 0–35 and a mean total of 20.46 (SD = 6.14; min. 7, max. 35) were obtained. 

Regarding the “parenting tasks” subscale, with a range of 0–40, a mean total of 20.14 (SD = 6.51; min. 7, max. 40) was obtained. These subscales may be useful in indicating how the parent/caregiver sees the situation, whether difficulties lie in the troublesome behavior of the children or the burden of meeting the ‘expected’ or ‘legitimate’ needs of the children. 

### 3.5. Global Lockdown Impact on the Relationship with the Children

Regarding the impacts of the lockdown on the relationships between the children and parents (as perceived by the parents), 54.1% (*n* = 1020) said it improved, 41.3% (*n* = 778) noted that it remained similar to the period before the lockdown, and 4.6% (*n* = 87) claimed it became worse.

Regarding parental perception of the positive developments of their children during the lockdown period, 97.3% (*n* = 1835) claimed that it did occur; of those, 37.8% (*n* = 712) believed that those developments would probably not have happened if it were not for the lockdown. In our sample, 14.7% (*n* = 277) of participants claimed that there was regression in their child’s development.

Compared with the period before the lockdown, 43.7% (*n* = 823) of participants claimed that their children had more defiant attitudes and behaviors, and were harder to manage.

Regarding the level of interference in caregivers carrying out their daily tasks (due to the child’s behavior), 45.8% (*n* = 864) of participants claimed that it was harder to accomplish their tasks without interference/interruptions. 

Finally, when participants were questioned whether their relational experiences with their children during the lockdown would promote any future changes in their family routines, 63.7% (*n* = 1200) claimed that it would, while 36.3% (*n* = 685) mentioned that it would not.

### 3.6. Univariate and Multivariate Analyses

The univariate analysis results corresponding to the three outcomes that defined the COVID-19 lockdown’s impact on relations between Portuguese parents and children from 1 to 3 years old are presented in Appendix A. 

### 3.7. Dependent Variable: Impact Perceived by Parents

Results of the multivariate analysis (Table 2) showed that parents who experienced positive changes in their child’s development during confinement were about four times (OR = 4.28; 95% CI: 2.05–8.91) more likely to have felt a positive impact on the relationship. On the contrary, parents who reported regressions in their child’s development were about 44% less likely to have felt a positive impact on the relationship (OR = 0.56; 95% CI: 0.33–0.93).

The multivariate model also showed that, for each unit increase in the total score of the challenging behavior subscale, there was a 12% decrease in the odds of a positive impact on the relationship between the caregiver and child (OR = 0.88; 95% CI: 0.85–0.92).

Interestingly, whether parents found it easier or more difficult to carry out their tasks (considering the interference from children in carrying them out), we can observe that these interferences were detrimental for a positive impact on the relationship between the two. Even so, without statistical significance, parents who found it easier to carry out their tasks were 29% less likely to report a positive impact on the relationship than before this period (OR = 0.71; 95% CI: 0.27–1.86); while those who found it more difficult to carry out their tasks were about 70% less likely to report a positive impact on the relationship (OR = 0.30; 95% CI: 0.16–0.59). 

### 3.8. Dependent Variable: Regressions in Children’s Development as Perceived by Parents

Regarding the “regressions in development” outcome, some independent variables with statistically significant associations were found.

Those who reported that there was a positive impact on the relationship were 51% less likely to have experienced regressions in development (OR = 0.49; 95% CI: 0.30–0.80).

The age of the child in confinement was also an important factor. As this increased the odds of regression felt by the parents, it became progressively greater. Parents of 2-year-old children, compared to 1-year-old children, were two-fold more likely to experience regression (OR = 2.20; 95% CI: 1.49–3.26); in the case of a 3-year-old child, the odds of having experienced regression was almost three times greater (OR = 2.87; 95% CI: 1.94–4.25).

Regarding the behavior of children after confinement, for parents who felt it was more collaborative and easier to manage, a 9% increase in the odds of experiencing regression was estimated (without statistical significance) when compared to the group of children who did not change their behavior (OR = 1.09; 95% CI: 0.68–1.73). However, for those who claimed that their children were more challenging/difficult to manage, the odds were already about three times greater (OR = 3.09: 95% CI: 2.20–4.36).

Finally, in PDHS, total scores above 70 on the intensity scale indicate that the parent is experiencing significant pressure in parenting. The results showed that parents who reported higher levels of pressure were two-fold more likely to feel regression compared to those who reported low pressure (OR = 2.22; 95% CI: 1.58–3.12).

### 3.9. Dependent Variable: Willingness to Promote Some Changes in Family Routines in the Future

Regarding sociodemographic factors, it was found that the current or previous follow-ups in psychiatry or psychology consultations were associated with the willingness for change. When compared to parents who have never had a follow-up, parents who had a follow-up were about 58% more likely to want to promote some change in their family routine in the future (OR = 1.58; 95% CI: 1.02–2.46). Regarding parents who had a follow-up in the past, weak evidence of an increase of about 23% in the odds of promoting changes was estimated (OR = 1.23; 95% CI: 0.98–1.53).

Concerning experiences during confinement, the number of adults who lived with the children was an important factor in this outcome. It was found that, for each adult who added to the household, there was a 24% increase in the odds of parents promoting changes in future family routines (OR = 1.24: 95% CI: 1.05–1-46). Still, in this context, we also concluded that parents who shared new experiences with their children during confinement were about three times more likely to consider changes in future family routines compared to those who did not share new experiences with their children (OR = 3.05; 95% CI: 2.11–4.42).

As previously mentioned, in PDHS, total scores above 50 on the frequency scale indicate that the parent is experiencing significant stress in parenting. Results showed that for parents who reported higher levels of stress, there was an increase of 40% in the odds of wanting future changes when compared to those who reported low stress (OR = 1.40; 95% CI: 1.12–1.76).

Finally, concerning the child’s behavior, parents who felt that the child was more collaborative and easier to manage after confinement were 2.81 times more prone to change when compared to those who had not felt changes in the child’s behavior (OR = 2.81; 95% CI: 2.12–3.73); for parents who claimed that the child was more challenging/difficult to manage, a 50% increase in the willingness to promote change was estimated (OR = 1.50; 95% CI: 1.21–1.87).

## 4. Discussion

For most families, the lockdown experience during the first wave of COVID-19 was a completely new and unexpected situation. Initially, urgent containment measures were essential to contain the spread of the virus, but, as Dr. Peter Green wrote in April 2020, “a shift in focus is needed to avoid an irreversible scarring of a generation” [18] (p. m1669). As the COVID-19 pandemic entered its second year, concerns regarding the lockdown impacts on specific groups of populations began to increase [19]. 

Although COVID-19 is seen as a global adverse event, the characteristics of lockdown strategies have varied across countries [12]. Personal experiences of lockdowns also varied widely depending on individual and family circumstances [20].

During the first wave, Portugal stood out as a country with greater control of the disease and decreased transmission rates after the application of severe containment measures. For example, schools at all education levels as well as childcare facilities remained closed from 16 March to 18 May 2020. At the same time, the Portuguese were witnessing the collapse of the Italian and Spanish health systems. The environment in which we lived was surrounded by great fear, motivating the population to comply with the recommended rules and to “stay at home”.

Some experts drew attention to the situation of confinement hypothetically putting children at risk: for instance, Fontanesi et al. argued that this context “increase the risk of trauma, including the loss of predictability in the known world, immobility, detachment or distancing, a lost sense of time and (…) security” [21] (p. S79). The same authors also warned about the following: “for low income parents and those with preexisting mental health problems, these challenges are likely to be exacerbated” [21] (p. S79). In the same sense, Peter Green warned about the following possibility: “for some families staying at home together increases the chances of child abuse or neglect” [18] (p. m1669), adding that “without regular access to professionals (…) routine opportunities to spot signs or narratives of safeguarding concern are lost” [18] (p. m1669).

In another direction, some works, such as by Herbert et al., have reinforced the hypothesis that “the effect of the pandemic is not uniformly or consistently negative across time for all individuals” [12] (p. 8). Some researchers have drawn attention to the possibility that there are opportunities for families to take positive aspects from this experience. In this sense, Cluver et al. claimed that “times of hardship can also allow for creative opportunity: to build stronger relationships with our children and adolescents” [22] (p. e64). The results in our study, in a way, can complement this more optimistic view. Note that, of the parents who answered the questionnaire, 95.4% (*n* = 1798) said that after confinement, the relationships with their children improved or remained similar to the pre-confinement period; 97.3% (*n* = 1835) observed changes in the child’s development and 63.7% (*n* = 1200) noted that, due to the relational experiences with the child during the confinement period, there will be some changes in family routines in the future.

Most sociodemographic variables related to parents (age, marital status, place of residence) did not allow us to make associations between these and the impact of confinement on families.

In this context, one exception that was observed as an important factor was the variable “current or previous follow-up in psychiatry or psychology consultations”. 

Parents who had follow-ups, compared to parents who never had follow-ups, showed more willingness to promote future changes in family routines. These data, which fall outside the focus of our study, appear interesting. In this sense, it might be interesting to attempt to assess the cause behind the desire to change family routines in this specific group of the population.

The child’s age, on the other hand, was another relevant sociodemographic factor in the perception of regression by the parents. It was observed that as the child’s age increased, the possibility of regression being felt by the parents was also progressively greater. These data can be explained by the trajectory of normal development in early childhood: while a 12-month-old child is even more focused on the dyadic relationship and the triangulation processes [23], for the 3-year-old child, the experience of socialization becomes more important; it may have been compromised during this period, which translated as a feeling of regression for the parents [24,25]. 

In contrast to the sociodemographic variables, which, as we have just described and with the two exceptions mentioned, are not particularly relevant, the variables that emerged with more associations to the outcomes were those that characterized aspects of the confinement period itself. Thus, it appears that the impact of the confinement period on the relationships between parents and children may have been more dependent on relational aspects, within the home, and on the parents’ sense of competence in exercising parental functions. For example, parents who reported regression in their child’s development were less likely to have felt a positive impact on their relationship with them. In this sense, it is understood why Perrin et al. argued that parenting is the best resource to reduce adverse impacts on a child’s social–emotional and behavioral development [26] (p. 637). 

These results are also in line with what was reported in recent studies carried out in other countries. Spinelli et al., on a sample involving Italian parents, concluded that “living in an at-risk contagion zone or being in closer contact with the virus had no significant effect on the well-being of parents and children” [4] (p. 4). Jiao et al. in China also concluded that the difference in a child’s symptoms (regarding areas identified by different levels of epidemic risks) was not statistically significant [13] (p. 265). Marchetti et al., in a study involving 1226 Italian parents, noted that “the experiences related to COVID-19 did not have a direct and significant impact on emotional exhaustion. Thus, pre-existing conditions and psychological variables may have a greater impact on the risk for developing parenting-related exhaustion than specific COVID-19 experiences” [27] (p. 1121). 

Contrary to what we initially believed, our results also revealed how parents developed their professional activities during the confinement did not seem to interfere significantly with the various outcomes. This was a surprise since we expected parents who suspended their professional obligations to report more positive impacts of the confinement experience on the relationships with their children. Following the same logic, we assumed that parents who worked remotely would be more overwhelmed and would have reported a more negative impact of the experience of confinement on the relationship with their children. Both assumptions were not directly observed.

The child’s behavior in general and the impact it had on the multiple tasks (professional or not) that the parents had to perform at home proved to be much more significant factors for the outcomes studied.

Therefore, although the perception of indicators of psychological issues was not one of the outcomes studied, it is not surprising that authors, such as Figueiredo et al., affirm that “whenever parents are able to stay close, to care, to love, to play, and to have conversations with their children, it is easier to perceive indicators of psychological issues and, then, interfere appropriately” [10] (p. 110176), regardless of whether the parents are working or not. All family members reciprocally influence each other’s adjustments and can help develop new resources by promoting positive adaptation during difficult times [28].

Contrary to the parents’ work situation, factors that were associated with the outcomes “regressions” and “willingness to change” in multivariate regression models were the significant levels of pressure over parenting and parental overload, reported by high scores in PDHS intensity and frequency scales.

Our results showed that parents who reported high levels of pressure were about twice as likely to experience regression when compared to those who reported low pressure. This is not surprising, as stated by Mikolajczak et al., “the mere fact of being a parent confronts one with a wide range of daily hassles, acute stressors, and even chronic stressors” [29] (p. 1319).

Therefore, it is also not surprising that parents who reported high levels of stress were more likely to desire future changes in the family routines compared to parents who reported low stress. Understandably, the first ones in the face of stress wanted to implement changes.

One relevant fact was that as the number of adults living with the child increased, the motivation for changes in family routines also increased. This factor possibly allowed parents to share the care, relieving the parental burdens and, consequently, their stress levels. In the same sense, Spinelli et al. concluded that “parents who report finding taking care of their children’s learning, finding space and time for themselves, the partner, the children, and for the activities they used to do before the lockdown” [4] (p. 5); they reported minor stress levels. As Spinelli et al. later claim: “it is the parents’ individual perception of the situation, and more specifically how difficult they find it dealing with the many stresses the quarantine imposes, that is significantly associated with parent’s stress and children’s psychological problems” [4] (p. 5).

Finally, having shared new experiences during the confinement period seems to have been a relevant factor in the impact on the parent–child relationship. In fact, 95.9% (*n* = 1675) considered it as having a positive impact on the relationship, while only 4.1% of caregivers considered it as having a negative impact. In the multivariate regression model, the “willingness to change” variable has a statistically significant association. In this case, parents who shared new experiences with their children during confinement were about three times more likely to consider future changes in family routines when compared to those who did not report new experiences with their children. Given these data, it seems right to say that, for many parents, the time spent with their children in confinement provided them with the opportunity to share different experiences from the usual ones, which can be seen as enriching in the relationship, motivating them to bring about changes in the future.

## 5. Conclusions

Our study can be seen as an important contribution to the literature concerning the most heard soundbite of the last year: the 2019 coronavirus pandemic is changing family life. The way family life has changed must now be the focus of our attention. In this sense, many studies have been conducted on the impacts of COVID-19 on the general population, particularly on the mental health of children and parents. Perhaps due to the inherent methodological difficulties, there are still few studies focused exclusively on the particular relationships between babies and their caregivers.

Thus, we carried out a cross-sectional study to evaluate the perception of the lockdown impact on relations between Portuguese parents and children between the ages of 1 and 3 during the COVID-19 pandemic. We concluded that in the population studied, despite the individual circumstances of each family and the increased demands imposed by confinement, a significant percentage of parents of children in this age group, positively evaluated the quality of their relationships with their children after this period. Variables, such as the child’s challenging behavior, interference in carrying out daily tasks (i.e., due to interference from children), and the parents’ perceptions of being under high pressure, negatively influenced the outcomes studied. On the contrary, parental perceptions concerning the development of their children and sharing new experiences with their children were significant factors in the positive impacts on the relationships between parents and their children, and on the desire to bring about changes in future family routines.

In this context, we agree with Griffiths et al. when they (quoting other authors) stated that “even at the best of times, when the world is not experiencing a pandemic, many parents experience stress specifically related to their roles as parents” [30] (p. 2). That being said, our data also highlight the resiliency of families and the bond that binds parents to their children between the ages of 1 and 3, and vice versa, even in times of adversity. Despite the levels of stress, risks, and fears, as we initially believed, this bond can be strengthened from the family experience of “being together” (as imposed during the first confinement). The same was concluded by Herbert et al., in a study involving 158 parents: “identifying positive aspects of the experience [of lockdown] may serve to buffer negative mental health risks across time. Understanding resilience strategies is critical for supporting current psychological wellbeing and to adequately prepare for future pandemic experiences” [12] (p. 1).

At the end of our study, participants were invited to leave comments. We’d like to highlight one that was representative of the majority of opinions expressed by the parents: “*My daughter started to crawl and ended up running, this being literally, but in some way also metaphorical. (…) The confinement was tiring, challenging but it was the best thing that happened to me/us. The management of the children, the house, the social isolation is tiring but, to be honest, I manage to do it very well and the girls were and are doing well.”* These results should encourage authorities to reflect on support policies for parents, encouraging them to look at the first years of a baby’s life in a more protective sense. To neglect this aspect is to run the risk that the following will come true (as stated by John Bowlby, in his lecture on caring for children over half a century ago): “Man and woman power devoted to the production of material goods counts a plus in all our economic indices. Man and woman power devoted to the production of happy, healthy, and self-reliant children in their own homes does not count at all. We have created a topsy-turvy world” [31] (p. 2).

In Mikolajczak et al.’s article on parental burnout, the authors quoted the first sentence from *A Tale of Two Cities* by Charles Dickens, noting that it could describe parenting. It is impossible to look at this period of lockdown and not find that the whole first paragraph applies: “It was the best of times, it was the worst of times, it was the age of wisdom, it was the age of foolishness, it was the epoch of belief, it was the epoch of incredulity, it was the season of light, it was the season of darkness, it was the spring of hope, it was the winter of despair, we had everything before us, we had nothing before us, we were all going direct to Heaven, we were all going direct the other way—in short, the period was so far like the present period, that some of its noisiest authorities insisted on its being received, for good or for evil, in the superlative degree of comparison only” [32] (p. 13). Children can be made more resilient, emotionally stronger, and better equipped to emerge out of the pandemic crisis toward a successful future [33,34,35].

## Figures and Tables

**Table 1 children-09-01124-t001:** Baseline characteristics of study participants.

Sociodemographic Characteristics *n* (%)	Participants (*n* = 1885)
Mother	1727 (91.6)
Father	158 (8.4)
Mean age (SD) (min, max)	35.5 (4.6) (20, 63)
*Marital status*	
Single	139 (7.4)
Married/couple	1746 (92.6)
*Education*	
Primary/Middle school	47 (2.5)
High school	207 (11)
College/university	1624 (86.2)
*Psychiatric history*	
Currently followed	110 (5.8)
Followed in the past	518 (27.5)
*Child age*	
1	531 (28.2)
2	740 (39.3)
3	614 (32.6)
Family routines before the COVID-19 lockdown *n* (%)	
Mean children in the household (SD) (min, max)	1.68 (0.9) (1, 10)
Mean adults in the household (SD) (min, max)	2.17 (0.6) (1, 7)
*Occupation*	
Employed	1746 (92.6)
Unemployed	81 (4.3)
*Who took care of the child*	
Kindergarten/babysitter	1573 (83.4)
Grandparents/other relatives	164 (8.7)
Mother/father/both	148 (7.9)
Family routines during the COVID-19 lockdown *n* (%)	
Stayed at the same address	1656 (87.9)
Maintained the same household composition	1700 (90.2)
*Working place*	
Worked from home	858 (45.5)
In the workplace	251 (13.3)
Laid off	466 (24.7)
*Who was taking care of the child*	
Single caregiver	510 (27.1)
Shared caregiving	1375 (72.9)
Sharing new experiences with the child	1747 (92.7)

Note: COVID-19, coronavirus disease 2019; min, minimum; max, maximum; SD, standard deviation.

**Table 2 children-09-01124-t002:** Results of multivariate regression models.

Model	OR-Estimate	95% CI	*p*-Value
Dependent variable: impact perceived by parents			
Evolution in the development of the child	4.28	2.05 to 8.91	<0.001
Regression in the child	0.56	0.33 to 0.93	0.026
Interference in daily tasks (due to the child’s behavior) ^a^			
Easier to handle tasks	0.71	0.27 to 1.86	0.479
Harder to handle tasks	0.30	0.16 to 0.59	<0.001
PDHS—subscale challenging behavior	0.88	0.85 to 0.92	<0.001
Dependent variable: regressions in the child’s development Perceived by parents			
Child’s age ^b^			
Two-year-old child	2.20	1.49 to 3.26	<0.001
Three-year-old child	2.87	1.94 to 4.25	<0.001
Positive impact on the relationship	0.49	0.30 to 0.80	0.004
Attitude and behavior of the child ^c^			
More challenging/harder to handle child	3.09	2.20 to 4.36	<0.001
More collaborative/easier to handle child	1.09	0.68 to 1.73	0.723
PDHS—high pressure on parents (>70) (IS)	2.22	1.58 to 3.12	<0.001
Dependent variable: willingness to promote some changes in family routines in the future			
Number of adults residing with the child in confinement	1.24	1.05 to 1.46	0.011
Psychiatric history ^d^			
Current psychiatric or psychologic follow-up	1.58	1.02 to 2.46	0.042
Past psychiatric or psychologic follow-up	1.23	0.98 to 1.53	0.073
Shared new experiences with the child	3.05	2.11 to 4.42	<0.001
Attitude and behavior of the child ^c^			
More challenging/harder to handle child	1.50	1.21 to 1.87	<0.001
More collaborative/easier to handle child	2.81	2.12 to 3.73	<0.001
PDHS—high stress of parents (>50) (FS)	1.40	1.12 to 1.76	0.004

Note: OR: odds ratio; CI: confidence interval; PDHS: Parenting Daily Hassles Scale; IS: intensity scale; FS: frequency scale. Reference categories: ^a^ there were no changes; ^b^ 1-year-old child; ^c^ there was no change in the child’s behavior; ^d^ no psychiatric history.

## Data Availability

The data that support the findings of this study are available from the corresponding author upon reasonable request.

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
