# Peer review of "The Lockdown Impact on the Relations between Portuguese Parents and Their 1- to 3-Year-Old Children during the COVID-19 Pandemic"

_children, 2022, doi:10.3390/children9081124_

Round 1

Reviewer 1 Report

The article addresses a set of factors that could have been associated with the impact perceived by parents, evolution in children development and willingness to promote some changes in family routines in the future in the context of COVID-19 pandemic. The importance of the study lies in the inclusion in the analysis of many socio-demographic factors, which creates an overview of families with young children in Portugal and their behaviors during the confinement period. Another strength of the study is the relatively high number of participants. The statistical analysis approach is appropriate, given that most independent variables are nominal. The study can attract readers' attention through attention to details and can contribute to the literature through explanations of the quality of parent-child relationships in critical socio-historical situations.

Some citations from Discussions section do not seem to conform to the style of the journal.

Line 179, 185 and entire article – Replace univariable and multivariable with univariate and multivariate

Line 190 - SPSS is named Statistical Package for Social Sciences

Minor revision in text editing is needed.

Author Response

Point 1Some citations from Discussions section do not seem to conform to the style of the journal.

Response 1 – We reviewed all the references in the article, changing those that were not in accordance with the style of the journal. All revisions are identified in red.

Point 2 - Replace univariable and multivariable with univariate and multivariate

Response 2 – we have changed the terms suggested by the reviewer. All changed terms are identified in red.

Point 3 - Line 190 - SPSS is named Statistical Package for Social Sciences

Response 3SPSS is indeed "named Statistical Package for Social Sciences" as stated by the reviewer, we changed the text accordingly.

Reviewer 2 Report

The paper focuses on the lockdown impact on relations between parents with their babies. The paper is interesting and original. 

I have only some minor suggestions:

- Introduction: add some more literature on COVID19 impact on adults' mental health or quality of life, for example the following: Bonichini, S., Tremolada, M. (2021). Quality of Life and Symptoms of PTSD during the COVID-19 Lockdown in Italy. Int. J. Environ. Res. Public Health, 18, 4385. https://doi.org/10.3390/ijerph18084385. 

Methodology/study design

It could be useful to have information also on Job hours and type of jobs to understand better the family situation and workload.

- Also the social support network is an important information to know before the lockdown effect: There was an important support or not before the pandemia, from who?

Discussion

Insert in the discussion selected studies that could avvalorate or confirm your results.

Author Response

Point 1 – Introduction: add some more literature on COVID19 impact on adults' mental health or quality of life.

Response 1 – As suggested, we have introduced a reference (Line 62) to the article: Bonichini, S., Tremolada, M. (2021). Quality of Life and Symptoms of PTSD during the COVID-19 Lockdown in Italy. Int. J. Environ. Res. Public Health, 18, 4385. https://doi.org/10.3390/ijerph18084385. 

Point 2 - It could be useful to have information also on Job hours and type of jobs to understand better the family situation and workload.

Response 2 – During the data collection we did not ask the parents about their type of job and we are unable to do it now. That said, we collected the data about whether the parents were unemployed before the lockdown and what was their working place.

Even though we did not ask the parents about their job hours we did collect data regarding waking up hours, bedtime and picking up from day-care schedules, we this in mind we estimated the average of hours parents spent with their children before the lockdown. That data is presented in the results and summarized again in Table 1.

Point 3 - Also the social support network is an important information to know before the lockdown effect: There was an important support or not before the pandemia, from who?

Response 3 – We agree and we did collect the most relevant data about the social support network as suggested by the reviewer. For instance, we asked about who took care of the child before the lockdown, whether the child was in a day care or with a nanny or wether she was taken care by their grandparents, other relatives, by their own mother, father or both. 

Point 4 - Insert in the discussion selected studies that could valorate or confirm your results.

Response 4 – As suggested by the reviewer we added to the “discussion” the following references:

Line 434 - Cusinato, M.; Iannattone, S.; Spoto, A.; Poli, M.; Moretti, C.; Gatta, M.; Miscioscia, M. Stress, Resilience, and Well-Being in Italian Children and Their Parents during the COVID-19 Pandemic. Int. J. Environ. Res. Public Health 2020, 17, 8297

and

Line 527 - Johnson, B. Importance of Positive Parenting During the Pandemic. BMH Medical Journal 2020 v. 7, n. 3, p. 46-49

 that may valorate or confirm our results.
